# An extensive program of periodic alternative splicing linked to cell cycle progression

Daniel Dominguez[1,2], Yi-Hsuan Tsai[1,3], Robert Weatheritt[4], Yang Wang[1,2], Benjamin J Blencowe[4]*, Zefeng Wang[1,5]*

[1]Department of Pharmacology, University of North Carolina at Chapel Hill, Chapel Hill, United States; [2]Lineberger Comprehensive Cancer Center, University of North Carolina at Chapel Hill, Chapel Hill, United States; [3]Program in Bioinformatics and Computational Biology, University of North Carolina at Chapel Hill, Chapel Hill, United States; [4]Donnelly Centre and Department of Molecular Genetics, University of Toronto, Toronto, Canada; [5]Key Lab of Computational Biology, CAS-MPG Partner Institute for Computational Biology, Chinese Academy of Science, Shanghai, China

**Abstract** Progression through the mitotic cell cycle requires periodic regulation of gene function at the levels of transcription, translation, protein-protein interactions, post-translational modification and degradation. However, the role of alternative splicing (AS) in the temporal control of cell cycle is not well understood. By sequencing the human transcriptome through two continuous cell cycles, we identify ~1300 genes with cell cycle-dependent AS changes. These genes are significantly enriched in functions linked to cell cycle control, yet they do not significantly overlap genes subject to periodic changes in steady-state transcript levels. Many of the periodically spliced genes are controlled by the SR protein kinase CLK1, whose level undergoes cell cycle-dependent fluctuations via an auto-inhibitory circuit. Disruption of CLK1 causes pleiotropic cell cycle defects and loss of proliferation, whereas CLK1 over-expression is associated with various cancers. These results thus reveal a large program of CLK1-regulated periodic AS intimately associated with cell cycle control.

*For correspondence:
b.blencowe@utoronto.ca (BJB);
zefeng@med.unc.edu (ZW)

## Introduction

Alternative splicing (AS) is a critical step of gene regulation that greatly expands proteomic diversity. Nearly all (>90%) human genes undergo AS and a substantial fraction of the resulting isoforms are thought to have distinct functions (*Pan et al., 2008*; *Wang et al., 2008*). AS is tightly controlled, and its mis-regulation is a common cause of human diseases (*Wang and Cooper, 2007*). Generally, AS is regulated by *cis*-acting splicing regulatory elements that recruit *trans*-acting splicing factors to promote or inhibit splicing (*Matera and Wang, 2014*; *Wang and Burge, 2008*). Alterations in splicing factor expression have been observed in many cancers and are thought to activate cancer-specific splicing programs that control cell cycle progression, cellular proliferation and migration (*David and Manley, 2010*; *Oltean and Bates, 2014*). Consistent with these findings, several splicing factors function as oncogenes or tumor suppressors (*Karni et al., 2007*; *Wang et al., 2014*), and cancer-specific splicing alterations often affect genes that function in cell cycle control (*Tsai et al., 2015*).

Progression through the mitotic cell cycle requires periodic regulation of gene function that is primarily achieved through coordination of protein levels with specific cell cycle stages (*Harashima et al., 2013*; *Vermeulen et al., 2003*). This temporal coordination enables timely control

**eLife digest** Mitosis is a key step in the normal life cycle of a cell, during which one cell divides into two new cells. As a cell progresses through the cell cycle, it must carefully regulate its gene activity to switch particular genes on or off at specific moments. When a gene is activated its sequence is first copied into a temporary molecule called a transcript. These transcripts are then edited to form templates to build proteins. One way that a transcript can be edited is via a process called alternative splicing, in which different pieces of the transcript are cut and pasted together to form different versions of the final template. This allows different instructions to be obtained from a single gene, introducing an added layer of biological complexity. However, the role of alternative splicing in the timing of key events of the cell life cycle is not well understood.

Dominguez et al. have now looked for the genes that undergo alternative splicing during the cell cycle. The sequences of gene transcripts produced within human cells were collected while the cells went through two rounds of division. This approach revealed that around 1,300 genes are spliced in different ways at different stages of each cell cycle. Many of these genes were known to play roles in controlling the cell's life cycle, but few of the genes showed large changes in the amount of total transcript that is generated over time.

Dominguez et al. also showed that an enzyme called CLK1 influences about half of the 1,300 periodically spliced genes during the cell cycle. The production of CLK1 is itself carefully controlled throughout the cell cycle, and the enzyme's activity prevents its own overproduction. Further experiments showed that blocking CLK1's activity while a cell is replicating its DNA halts the cell cycle, but blocking this enzyme's activity after the cell had replicated its DNA did not. Given this pivotal role in the cell cycle, Dominguez et al. also examined the role of CLK1 in cancer cells and found that high levels of CLK1 in tumours were linked to lower survival rates. These findings indicate that CLK1 warrants further investigation, particularly in relation to its role in cancer.

of molecular events that ensure accurate chromatin duplication and daughter cell segregation. Periodic gene function is conventionally thought to be achieved through stage-dependent gene transcription (*Bertoli et al., 2013*), translation (*Grabek et al., 2015*), protein-protein interactions (*Satyanarayana and Kaldis, 2009*), post-translational protein modifications, and ubiquitin-dependent protein degradation (*Mocciaro and Rape, 2012*). Although AS is one of the most widespread mechanisms involved in gene regulation, the relationship between the global coordination of AS and the cell cycle has not been investigated.

Major families of splicing factors include the Serine-Arginine rich proteins (SR) proteins and the heterogeneous nuclear ribonucleoproteins (hnRNPs), whose levels and activities vary across cell types. SR proteins generally contain one or two RNA recognition motifs (RRMs) and a domain rich in alternating Arg and Ser residues (RS domain). Generally, RRM domains confer RNA binding specificity while the RS domain mediates protein-protein and protein-RNA interactions to affect splicing (*Long and Caceres, 2009*; *Zhou and Fu, 2013*). Post-translational modifications of SR proteins, most notably phosphorylation, modulate their splicing regulatory capacity by altering protein localization, stability or activity (*Gui et al., 1994*; *Lai et al., 2003*; *Prasad et al., 1999*; *Shin and Manley, 2002*). Dynamic changes in SR protein phosphorylation have been detected after DNA damage (*Edmond et al., 2011*; *Leva et al., 2012*) and during the cell cycle (*Gui et al., 1994*; *Shin and Manley, 2002*), suggesting that regulation of AS may have important roles in cell cycle control. However, the functional consequences of SR protein (de)phosphorylation during the cell cycle are largely unclear.

Through a global-scale analysis of the human transcriptome at single-nucleotide resolution through two continuous cell cycles, we have identified widespread periodic changes in AS that are coordinated with specific stages of the cell cycle. These periodic AS events belong to a set of genes that is largely separate from the set of genes periodically regulated during the cell cycle at the transcript level, yet the AS regulated set is significantly enriched in cell cycle- associated functions. We further demonstrate that a significant fraction of the periodic AS events is regulated by the SR protein kinase, CLK1, and that CLK itself is also subject to cell cycle-dependent regulation. Moreover,

inhibition or depletion of CLK1 causes pleiotropic defects in mitosis that lead to cell death or G1/S arrest, suggesting that the temporal regulation of splicing by CLK1 is critical for cell cycle progression. The discovery of periodic AS thus reveals a widespread yet previously underappreciated mechanism for the regulation of gene function during the cell cycle.

## Results

### Alternative splicing is coordinated with different cell cycle phases

To systematically investigate the regulation of AS during the cell cycle, we performed an RNA-Seq analysis of synchronously dividing cells using a total of 2.3 billion reads generated across all stages (G1, S, G2 and M) of two complete rounds of the cell cycle (*Figure 1—figure supplement 1A*). To maximize the detection of regulated AS events, we used the complementary analysis pipelines, MISO and VAST-TOOLS (*Katz et al., 2010*; *Irimia et al., 2014*). These pipelines have different detection specificities and employ partially overlapping reference sets of annotated AS events, and therefore afford a more comprehensive analysis when employed together. Both pipelines were used to determine PSI (the percent of transcript with an exon spliced in) and PIR (the percent of transcripts with an intron retained). Alternative exons detected by both pipelines had highly correlated PSI values (*Figure 1—figure supplement 1G*; see below). Consistent with previous results (*Bar-Joseph et al., 2008*; *Whitfield et al., 2002*), transcripts from approximately 14.2% (1182) of expressed genes displayed periodic differences in steady-state levels between two or more cell cycle stages (see below). Remarkably, 15.6% (1293) of expressed genes also contained 1747 periodically-regulated AS events, among a total of ~40,000 detected splicing events (FDR < 2.5%; *Figure 1A* and *Figure 1—figure supplement 1B,D*).

Importantly, as has been observed previously for AS regulatory networks (*Pan et al., 2004*), the majority of genes with periodic AS events did not overlap those with periodic steady-state changes in mRNA expression. This indicates that genes with periodic changes in AS and transcript levels are largely independently regulated during the cell cycle (*Figure 1B*). Further supporting this conclusion, we did not observe a significant correlation (positive or negative) between exon PSI values and mRNA expression levels for genes with both periodic expression and periodic exon skipping (data not shown). A gene ontology (GO) analysis reveals that genes with periodic AS, like those with periodic transcript level changes (*Bar-Joseph et al., 2008*; *Whitfield et al., 2002*), are significantly enriched in cell cycle-related functional categories, including M-phase, nuclear division and DNA metabolic process (*Figure 1C*; adjusted p<0.05 for all listed categories, FDR<10%) (*Supplementary file 1*). Similar GO enrichment results were observed when removing the relatively small fraction (10%) of periodically spliced genes that also display significant mRNA expression changes across the cell cycle (*Figure 1C*). These results thus reveal that numerous genes not previously linked to the cell cycle, as well as previously defined cell cycle-associated genes thought to be constantly expressed across the cell cycle, are in fact subject to periodic regulation at the level of AS (*Supplementary file 1* for a full list).

Among the different classes of AS analyzed (cassette exons, alternative 5'/3' splice sites and intron retention [IR]), periodically regulated IR events were over-represented (relative to the background frequency of annotated IR events) by ~2.2 fold whereas periodically regulated cassette exons, represent the next most frequent periodic class of AS (p=2.2×10⁻¹⁶, Fisher's exact test, *Figure 1—figure supplement 1E*). Quantitative RT-PCR assays across different cell cycle stages validated periodic IR events detected by RNA-Seq (*Figure 1D*). Interestingly, one of these IR events is in transcripts encoding aurora kinase B (AURKB), a critical mitotic factor regulated at the levels of transcription, protein localization, phosphorylation and ubiquitination (*Carmena et al., 2012*; *Lens et al., 2010*). The AURKB retained intron is predicted to introduce a premature termination codon that elicits mRNA degradation through nonsense mediated decay, and is thus expected to result in reduced levels of AURKB protein. The splicing of the retained intron lags behind changes in the total *AURKB* mRNA expression (*Figure 1E*). We computationally corrected levels of fully spliced, protein coding *AURKB* mRNA by taking into account the fraction of intron-retaining (i.e. non-productive) transcripts across the cell cycle stages (*Figure 1E*). The expression curve for corrected *AURKB* mRNA levels is substantially different from total *AURKB* transcript levels, with a shifted peak coinciding with mitosis. Periodically-regulated IR events detected in other genes, including those with

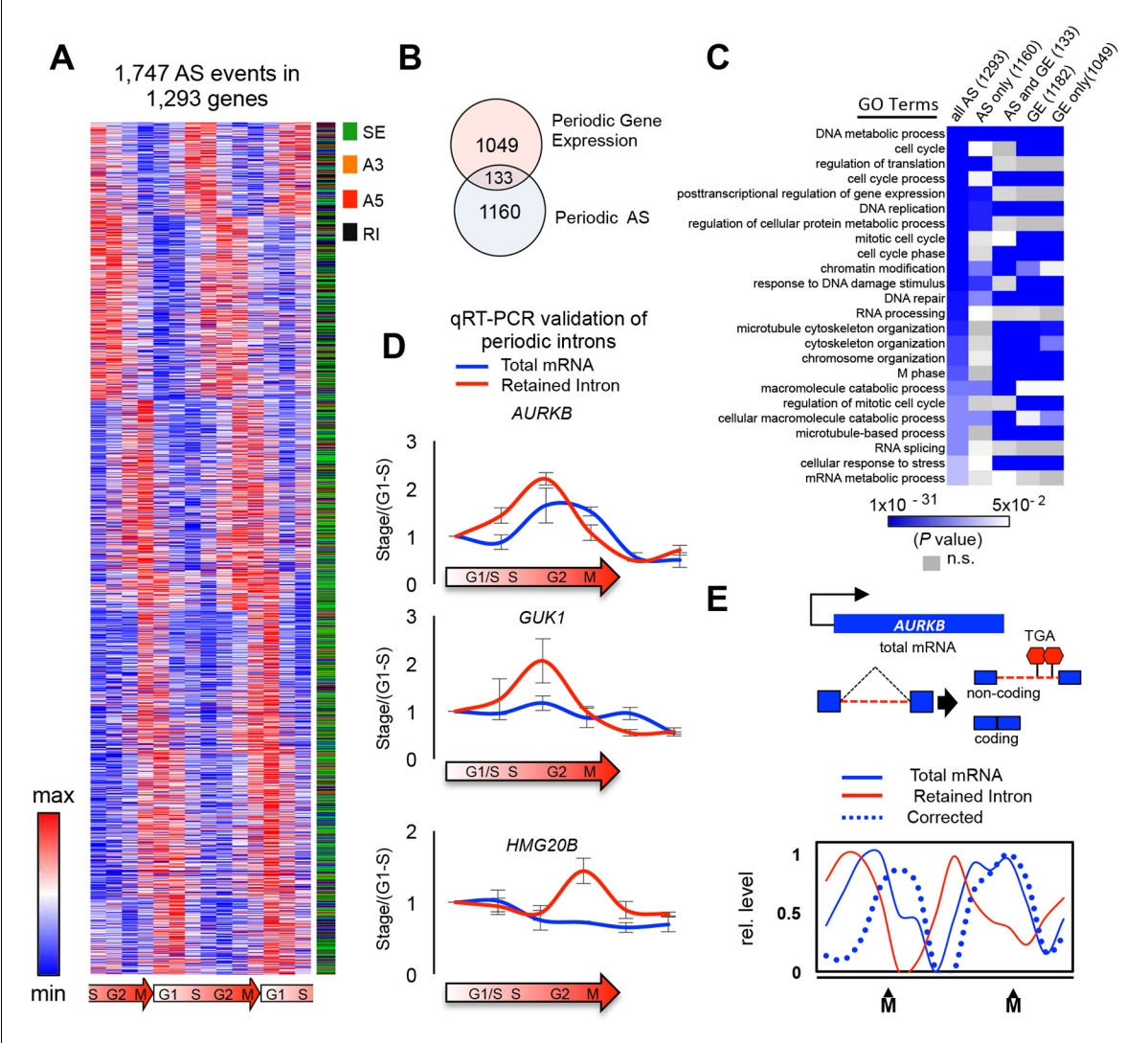

**Figure 1.** Global detection of periodic cell cycle-dependent alternative splicing. (**A**) Heat map representation of periodically spliced events. Row-normalized relative PSI values are shown. Diagram below indicates cell cycle phase. (**B**) Overlap between periodically spliced genes and periodically expressed genes detected by RNA-Seq. (**C**) Heat map representation of enriched Gene Ontology terms shown as log (p-value). Three gene sets were analyzed separately: all genes with periodic AS, genes with periodic AS only, and genes with both periodic AS and periodic expression. (**D**) Real-time quantitative PCR analysis of periodic retained introns and total mRNAs for three selected genes. Cells were synchronized by double thymidine block and samples were collected 0, 3, 6, 9, 12 and 15 hr post release. Errors bars represent standard deviation of the mean. Diagram below indicates cell cycle stage. (**E**) Schematic representation of *AURKB* AS pattern. Line graph showing the relationship between intron retention and mRNA levels for the *AURKB* gene across the cell cycle. Percent intron retention (solid red line) across cell cycle was used to determine the fraction of total mRNAs (solid blue line) not containing an intron, i.e. 'corrected' mRNA levels (dashed blue line).

The following figure supplement is available for figure 1:

**Figure supplement 1.** Identification of periodic AS by multiple analysis pipelines.

known cell cycle functions such as HMG20B and RAD52, are similarly expected to affect the cell cycle timing of mRNA expression (***Figure 1A,D***). Collectively, these results provide evidence that the temporal control of retained intron AS provides an important mechanism for establishing the timing of expression of AURKB mRNA and protein, as well as of the timing of expression of additional genes during the cell cycle.

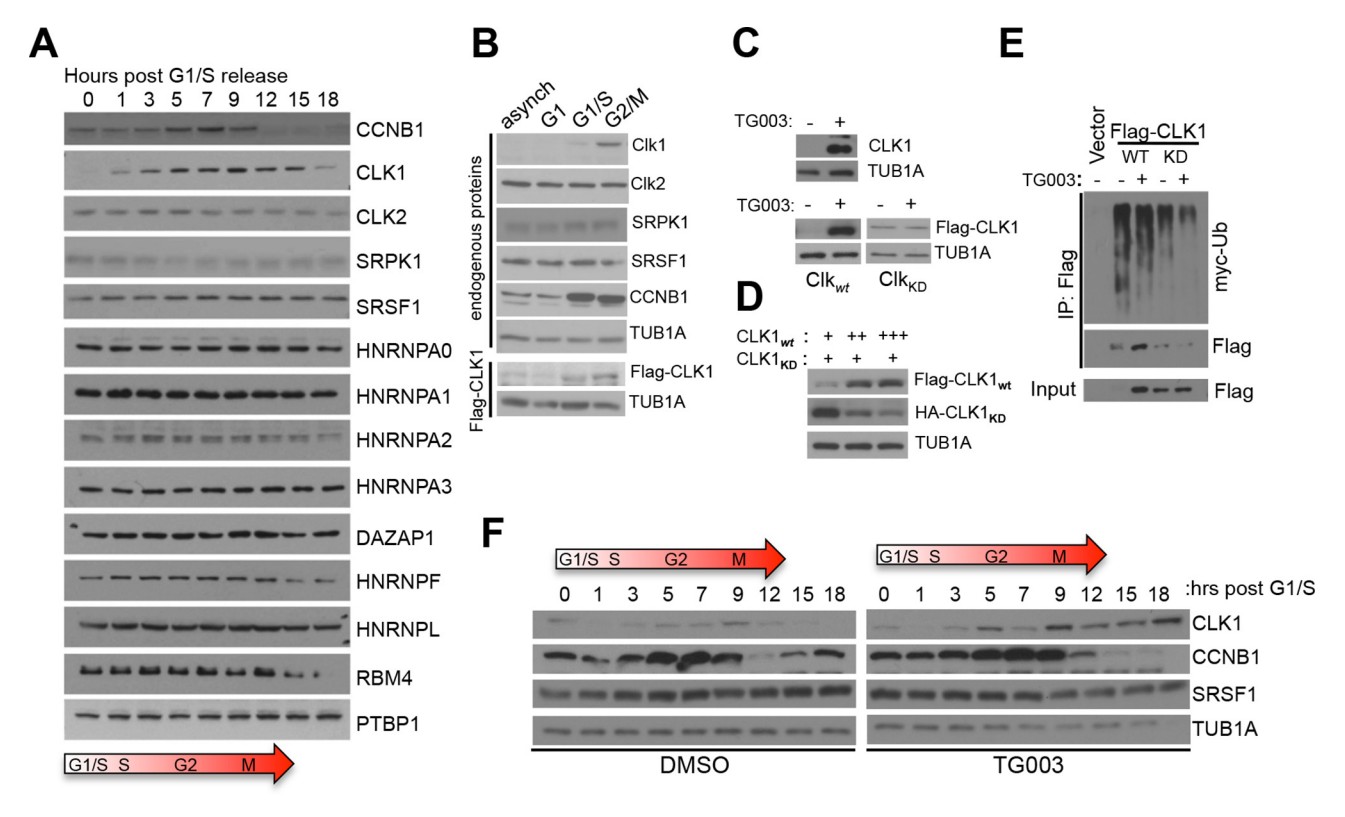

**Figure 2.** Cell cycle-dependent regulation of CLK1. (**A**) Immunoblot analysis of proteins involved in splicing regulation in synchronized HeLa cells after release from double thymidine block. (**B**) Immunoblot analysis of selected proteins in asynchronous HeLa cells or cells arrested at different cell cycle stages. Stably expressed exogenous CLK1 levels were also assessed during the cell cycle (bottom panel). (**C**) Immunoblot of endogenous CLK1 (top) and exogenously-expressed wild type (CLK1$_{wt}$) or kinase catalytically inactive (CLK1$_{KD}$) proteins (bottom) upon treatment with 10 μM TG003. (**D**) Co-expression of CLK1$_{WT}$ and CLK1$_{KD}$ at different ratios. (**E**) Immunoprecipitation of CLK1 proteins co-expressed with myc-ubiquitin. Cells were treated with 10 μM TG003 and 10 μM MG132 prior to sample collection. (**F**) Immunoblot analysis of lysates from cells synchronized upon early S phase (double thymidine) release with or without TG003 treatment.

The following figure supplements are available for figure 2:

**Figure supplement 1.** Periodic expression of RBPs.

**Figure supplement 2.** Regulation of CLK1 proteins levels during the cell cycle is degradation-dependent.

## The SR protein kinase CLK1 fluctuates during the cell cycle

Alternative splicing is generally regulated by the concerted action of multiple *cis*-elements that recruit cognate splicing factors. Consistently, analysis of our RNA-seq data revealed 96 RNA binding proteins (RBPs) with periodic mRNA expression, including RS domain-containing factors like *SRSF2, SRSF8, TRA2A* and *SRSF6* (*Figure 2—figure supplement 1A*). These 96 RBPs were significantly enriched in the GO term 'splicing regulation' (adjusted $p=10^{-4}$, *Figure 2—figure supplement 1B*), indicating that periodic AS is likely controlled by multiple RBPs. Correlations between these RBPs and periodic splicing events were also identified (*Figure 2—figure supplement 1C*). For example, SRSF2 expression is significantly correlated with the splicing pattern of a retained intron in the *SRSF2* transcript. Further supporting a role for these RBPs in controlling periodic splicing was the identification of RNA motifs bound by a subset of periodically expressed RBPs (*Figure 2—figure supplement 1D*). To further examine periodic RBP regulation during cell cycle, we measured the abundance of known splicing regulatory proteins at different stages of the cell cycle by immunoblotting (*Figure 2A*). Among the proteins analyzed, CDC-like kinase 1 (CLK1), an important regulator of the Ser/Arg (SR) repeat family of splicing regulators, displayed the strongest cyclic expression

peaking at the G2/M phase (*Figure 2A,B*), consistent with the results of a recent mass-spectrometry-based screen for cycling proteins (*Ly et al., 2014*). CLK1 is one of four human CLK paralogs (CLK1-4) and is known to regulate AS via altering the phosphorylation status of multiple SR proteins (*Duncan et al., 1997*; *Jiang et al., 2009*; *Ninomiya et al., 2011*; *Prasad et al., 1999*). Notably, the levels of other detectable CLK paralogs, as well as members of another SR protein kinase, SRPK1, did not change significantly at the level of RNA and/or protein during the cell cycle (*Figure 2B* and *Supplementary file 1*).

Given that both CLK1 protein levels and known CLK1 substrates are periodically expressed, we decided to further investigate the role of CLK1 in the context of cell cycle. The levels of total CLK1 mRNA, as well as the levels of specific CLK1 splice variants, did not change significantly during the cell cycle (*Figure 2—figure supplement 2A,B*) indicating that periodic expression of CLK1 is controlled at the level of protein translation and/or turnover. Consistent with this, an exogenously expressed CLK1 protein displayed cell cycle-dependent fluctuations similar to those observed for endogenous CLK1 protein (*Figure 2B*). Moreover, CLK1 was rapidly degraded upon inhibition of translation by cycloheximide, and this effect was reversed by co-treatment with the proteasome inhibitor MG132 (*Figure 2—figure supplement 2C*). Additionally, polyubiquitination of Flag-tagged CLK1 was detected following immunoprecipitation with anti-Flag antibody from cells treated with MG132 (*Figure 2—figure supplement 2D*). These data suggest that the levels of CLK1 protein are controlled by ubiquitin-mediated degradation in a cell cycle-dependent manner.

Periodically regulated protein levels are often controlled through negative feedback circuits involving auto-regulatory loops. CLK1 has been reported to auto-phosphorylate on several residues (*Ben-David et al., 1991*). To investigate whether auto-phosphorylation of CLK1 affects its periodic regulation, we tested whether blocking its kinase activity affects its stability. Inhibition of CLK1 kinase activity using a selective inhibitor, TG003 (*Muraki et al., 2004*), markedly stabilizes both endogenous and exogenously expressed CLK1 proteins (*Figure 2C*). Moreover, activity-dependent destabilization of CLK1 was observed with a wild type (WT) protein, but not with a catalytically inactive (KD) mutant (*Figure 2C*, left panel). We further observe that WT CLK1 is rapidly degraded upon cycloheximide treatment, whereas the KD mutant is more stable (*Figure 2—figure supplement 2C*). We next tested whether CLK1 activity is sufficient to trigger its own degradation by co-expressing KD CLK1 with increasing amounts of WT CLK1. As expected, increasing amounts of WT CLK1 reduces levels of KD CLK1 (*Figure 2D*). Consistent with these results, WT CLK1 is more highly polyubiquitinated compared to the KD mutant (*Figure 2E*, compare lanes 2 to 4), and treatment with TG003 reduces polyubiquitination levels (*Figure 2E*, lane 2 *vs.* 3 and lane 4 *vs.* 5). Decreased polyubiquitination of WT CLK1 is more prevalent than is apparent upon TG003 treatment, as CLK1 is stabilized by TG003 inhibition and thus more total Flag-CLK1 is immunoprecipitated (*Figure 2E*). To further examine whether this auto-feedback loop is required for changes in CLK1 protein levels during the cell cycle, we treated synchronized cells with TG003 (or DMSO as a control) and measured CLK1 protein levels. We observed that CLK1 inhibition prevents its turnover after the G2/M phase for both endogenous and exogenously expressed kinases (*Figure 2F* and *Figure 2—figure supplement 2E*). Taken together, these results provide strong evidence that CLK1 protein levels are controlled by ubiquitin-mediated proteolysis in a cell cycle stage-specific manner, and that an activity-dependent negative feedback loop is required for this periodic regulation. These results further suggest that changes in the levels of CLK1 could account for many of the periodically regulated AS transitions we have detected during the cell cycle.

## CLK1 regulates AS events in genes with critical roles in cell cycle control

RNA-Seq analysis of cells treated with TG003 revealed 892 AS events (in 665 genes) that significantly change after CLK1 inhibition (*Figure 3A*), including known CLK1-regulated splicing events (e.g. exon 4 of CLK1 [*Duncan et al., 1997*]). It is worth noting that TG003 can also inhibit CLK4 (although to a lesser extent than inhibition of CLK1). However, RNAi of CLK1 is sufficient to recapitulate the phenotype of CLK1/4 inhibition (see below and *Figure 4*) (*Fedorov et al., 2011*; *Muraki et al., 2004*). Intron retention and cassette exons are the most overrepresented types of AS affected by TG003 (*Figure 3A*). Most (70%) of the CLK1-regulated exons display increased skipping upon CLK1 inhibition, whereas 87% of CLK1-regulated introns show increased retention (*Figure 3—figure supplement 1B and C*), consistent with a recently reported role for CLK1 in the regulation of retained

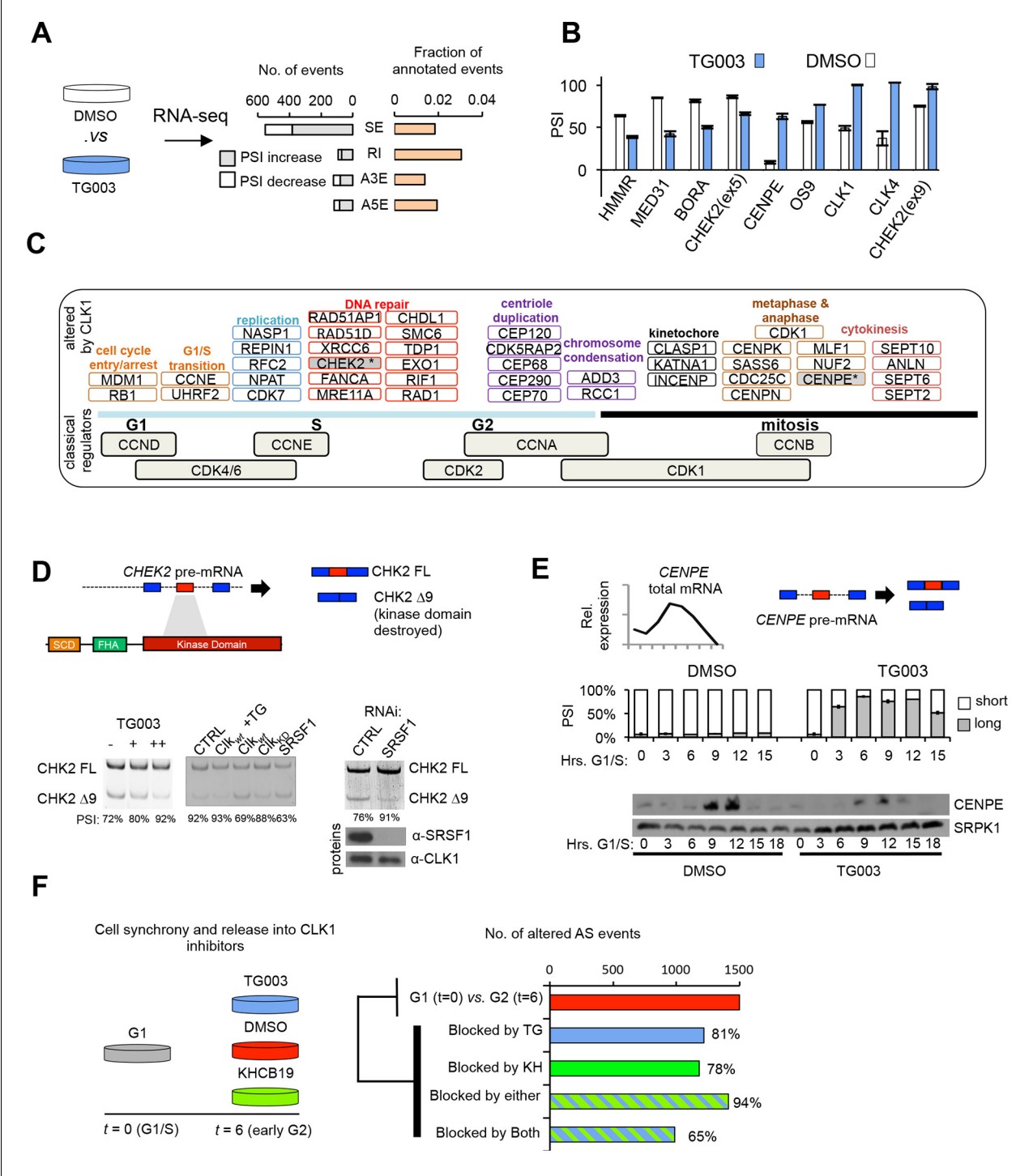

**Figure 3.** CLK1 regulates a network of genes that control cell cycle progression. (**A**) Identification of endogenous CLK1 targets by RNA-Seq. Numbers of different AS types affected by treatment with the CLK1 inhibitor TG003 (left graph). SE, skipped exon; RI, retained intron; A3E, alternative 3′ exon; A5E, alternative 5′ exon. Fraction of total analyzed events that were affected by TG003 treatment (right graph). (**B**) Validation of TG003-responsive AS events by semi-quantitative RT-PCR. The bar graph shows the max-delta PSI for each AS event tested in a 24-hr time course of inhibition with 20μM TG003. (**C**) Representation of cell cycle control genes with CLK1-dependent AS events, organized by cell cycle phase and function. (**D**) Schematic representation of CHEK2 alternative splicing, showing that exon 9 encodes a region overlapping the kinase domain (upper panel). Semi-quantitative RT-PCR assessment of CHEK2 isoforms after treatment with TG003 or over-expression of the indicated factors (lower left panel). RNAi of SRSF1 in cells and subsequent analysis of CHEK2 splicing by semi-quantitative RT-PCR (lower right panel). PSI values are shown below gel. (**E**) Normalized *CENPE* total mRNA expression during an unperturbed cell cycle (triangle denotes mitosis, left panel) and diagram of *CENPE* splicing (right). TG003-treatment

*Figure 3 continued on next page*

*Figure 3 continued*

of HeLa cells released from G1/S arrest followed by semi-quantitative RT-PCR analysis of CENPE isoforms (bar graph). (F) Schematic of RNA-Seq analysis of CLK1 inhibition during cell cycle (left). AS events that were identified as being differentially regulated between G1 and G2 phase (top bar of bar graph) and number of events that were blocked by the indicated conditions (bottom 4 bars).

The following figure supplement is available for figure 3:

**Figure supplement 1.** CLK1 regulates a network of genes that control cell cycle progression.

introns (*Boutz et al., 2015*). Of nine analyzed TG003-affected AS events detected by RNA-Seq analysis, all were validated by semi-quantitative RT-PCR assays (*Figure 3B*). These observations indicate that CLK1 inhibition mainly suppresses splicing, consistent with a general requirement for phosphorylation of SR proteins to promote splicing activity (*Irimia et al., 2014*; *Prasad et al., 1999*; *Tsai et al., 2015*). Importantly, there is a significant overlap between genes with cell cycle periodic AS events (*Figure 1*) and those with CLK1-regulated AS events, involving 156 genes (p=$8.5\times10^{-10}$, hyper-geometric test). In contrast, consistent with the results in *Figure 1*, we do not observe a significant overlap between genes containing CLK1-regulated AS events and periodically expressed genes. These results thus support a widespread and rapidly acting role for CLK1 in controlling cell cycle-regulated AS. Indeed, CLK1 inhibition induces rapid (within 3–6 hr) changes in AS among several analyzed cases (*Figure 3—figure supplement 1D*).

Supporting an important role for CLK1 in cell cycle progression, genes whose AS levels are affected by CLK1 inhibition are significantly enriched in the GO terms cell cycle phase, M-phase, DNA metabolic processes, nuclear division, DNA damage response, and cytokinesis (adjusted p<0.05 and FDR <20% for all listed GO terms; full list in *Supplementary file 2*). The affected genes function at various stages of cell cycle including the G1/S transition (*Figure 3C*). Mitotic processes were, however, associated with the largest number of CLK1-target genes with AS changes and included examples that function in centriole duplication (*CEP70, CEP120, CEP290, CEP68, CDK5RAP2*), metaphase and anaphase (e.g. *CENPK, CENPE, CENPN*), and cytokinesis (e.g. *SEPT2, SEPT10, ANLN*) (additional examples in *Figure 3C*).

To further investigate the functional consequence of CLK1-dependent AS, we selected two examples in genes that have important roles in the cell cycle: *checkpoint kinase 2 (CHEK2)*, a tumor suppressor that controls the cellular response to DNA damage and cell cycle entry (*Paronetto et al., 2011*; *Staalesen et al., 2004*), and *centromere-associated protein E (CENPE)*, a kinetochore-associated motor protein that functions in chromosome alignment and segregation during mitosis (*Kim et al., 2008*). We detected a TG003 dose-dependent increase in *CHEK2* exon 9 inclusion, whereas overexpression of WT CLK1 induced exon 9 skipping, an event that removes the CHEK2 kinase domain (*Figure 3D*). Expression of WT CLK1 in the presence of TG003, or a catalytically inactive CLK1, had little to no effect on the splicing of this exon (*Figure 3D*, bottom panels), indicating that the catalytic activity of CLK1 is essential for regulating CHEK2 AS. Over-expression of the SR protein splicing regulator SRSF1, a known target of CLK1 (*Prasad et al., 1999*), had a similar effect as over-expression of CLK1, resulting in CHEK2 exon 9 skipping, whereas knockdown of SRSF1 had the opposite effect (*Figure 3D*, right panel). Furthermore, the activation of CHEK2 requires homodimerization (*Shen et al., 2004*), and we observe that the CHEK2 isoform lacking exon 9 still interacts with full-length protein (*Figure 3—figure supplement 1E*), suggesting that this CLK1-regulated isoform may function in a dominant-negative manner to attenuate CHEK2 activity.

CENPE is known to be tightly controlled at multiple levels (including transcription, localization, phosphorylation and degradation), and disruption of its regulation leads to pronounced mitotic defects. *CENPE* AS generates long and short isoforms (*Supplementary file 2*), with the predominant variant being the short isoform that lacks amino acids 1972–2068. Inhibition of CLK1 rapidly shifts *CENPE* splicing to produce predominantly the long isoform (*Figure 3E*), and this is accompanied by a reduction in *CENPE* protein levels during G2/M phase, presumably due to the instability of the long isoform (*Figure 3E*). These data thus show that CLK1 controls the AS of major cell cycle regulators, and therefore suggest that inhibition of CLK1 may alter cell cycle progression.

To investigate this, we next performed an RNA-Seq analysis of synchronized cells at G1 and early G2 (when CLK1 accumulates), following inhibition of CLK1 with TG003. As a specificity control, we

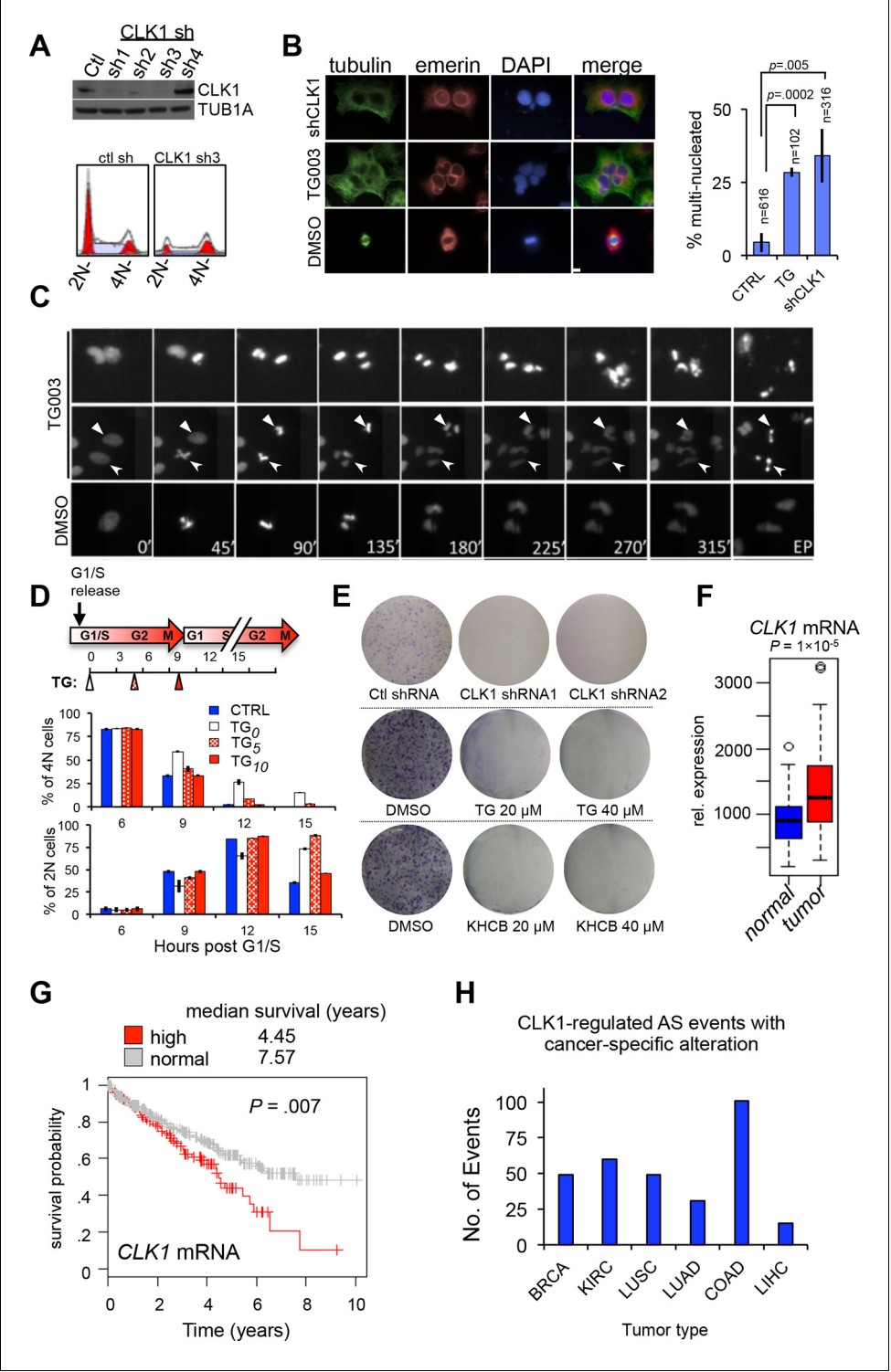

**Figure 4.** CLK1 is required for cell cycle progression and proliferation. (**A**) Immunoblot analysis of CLK1 proteins after stable shRNA knockdown in HeLa cells. Bottom, DNA content as measured by propidium iodide staining following flow cytometry. (**B**) Immunofluorescence microscopy of A549 cells depleted of CLK1 by shRNA (top row), cells treated with 10 μM TG003 for 12 hr (middle row), and a control treatment with DMSO (bottom row); green: tubulin, red: emerin (nuclear envelope), and blue: DAPI. Scale bar 10 μm. Right bar graph shows the quantification of multinucleated cells. p values determined using Student's t-test. (**C**) Static frames from a live-cell high-content imaging movie of HeLa cells expressing Histone H2B-GFP and treated with TG003 (top panel). Time after start of
*Figure 4 continued on next page*

*Figure 4 continued*

the experiment is indicated; EP, end point (~960 min). TG003 treated cells with apparent cell division defects (indicated by arrowheads in the bottom field) are shown in two independent fields. (D) Synchronized HeLa cells were treated with 20 μM TG003 at the indicated time points (0, 5, and 10 hr) and analyzed by propidium iodide staining and flow cytometry to measure DNA content. Percent of 2N (lower bar graph) and 4N (upper bar graph) cells were quantified at each time point as indicated in the treatment scheme (top). (E) Colony formation assay of HeLa cells depleted of CLK1 by shRNA, or continuously treated with TG003 or KHCB-19 at the indicated concentrations. (F) Box plot representation of CLK1 mRNA expression levels in paired normal and tumorous kidney tissue. 72 cases were analyzed. (G) Kaplan-Meier plot showing survival differences between patients with kidney tumors with high CLK1 (red, upper quartile) or reduced CLK1 (blue, lower three quartiles) expression. (H) Number of cancer-associated AS events that are also regulated by CLK1 in different tumor types. BRCA, Breast invasive carcinoma; COAD, Colorectal adenocarcinoma; KIRC, Kidney renal clear cell carcinoma; LUAD, Lung adenocarcinoma; LUSC, Lung squamous cell carcinoma; LIHC, liver hepatocellular carcinoma.

The following figure supplements are available for figure 4:

**Figure supplement 1.** Loss of CLK1 results in cell cycle defects in multiple cell types.

**Figure supplement 2.** CLK1 mis-regulation in human cancer.

---

performed a parallel RNA-Seq analysis using a structurally distinct CLK1 inhibitor, KHCB-19 (*Fedorov et al., 2011*). Strikingly, of 1498 AS events that change between G1 ($t$=0) and G2 ($t$=6) ~94% display reduced changes following treatment with the two drugs (*Figure 3F*), with 65% commonly affected by both drugs, thus supporting an important role for CLK1 in controlling cell-cycle dependent splicing.

## CLK1 is required for normal mitosis and cell proliferation

Given that CLK1 regulates the AS of many cell cycle factors (*Figure 3C*), we next examined whether it is necessary for cell cycle progression. Knockdown of CLK1 using shRNAs led to an accumulation of cells with 4N DNA content in multiple cell types, specifically HeLa, H157, and A549 (*Figure 4A*, *Figure 4—figure supplement 1A,B*), and the extent of this accumulation correlated with the degree of knockdown (*Figure 4—figure supplement 1A*). We also observed a significant increase in multi-nucleation, a common consequence of defective chromosome segregation or cytokinesis, following shRNA-knockdown or TG003 inhibition of CLK1 in the treated cells (*Figure 4B* and *Figure 4—figure supplement 1C*). To visualize the effect of CLK1 inhibition on mitosis at a single cell level, we performed time-lapse high-content microscopy on live cells stably expressing a GFP-histone 2B fusion protein, to track changes in chromatin. TG003-treated cells entered mitosis normally, as measured

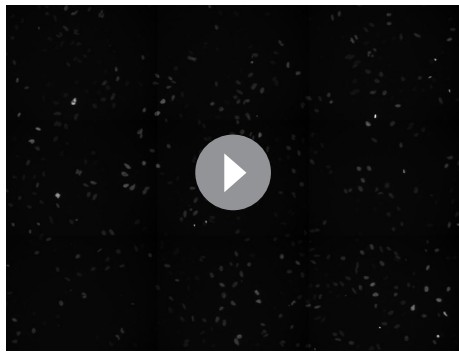

**Video 1.** Live-cell imaging of control HeLa cells stably expressing a GFP-H2B. Cells were synchronized by single-thymidine block and released and imaged at 10X magnification (every ~15 min) for 960 min.

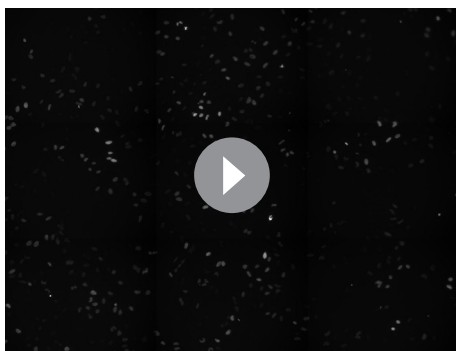

**Video 2.** Live-cell imaging of control HeLa cells stably expressing GFP-H2B. Cells were synchronized by single-thymidine block and released into 1 μM of TG003 and imaged at 10X magnification (every ~15 min) for 960 min.

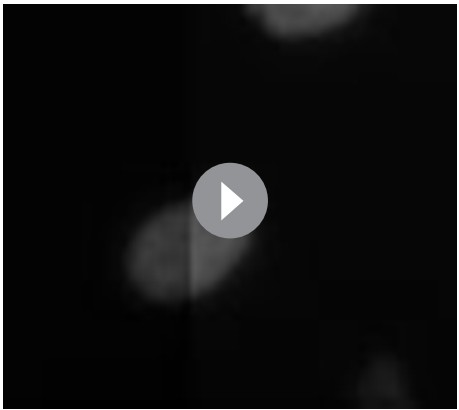

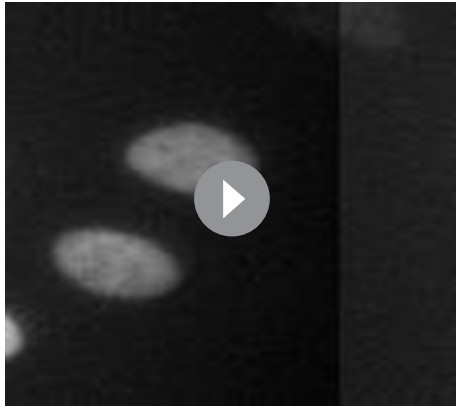

**Video 3.** Zoom of live-cell imaging of control HeLa cells. Data associated with *Figure 4D*.

**Video 4.** Zoom of live-cell imaging of TG003-treated HeLa cells. Data associated with *Figure 4*.

by nuclear envelope breakdown, but displayed delayed or aberrant cytokinesis, typically resulting in multi-polar divisions, increased time in metaphase, failure to undergo chromatin de-condensation and eventual cell death (*Figure 4C* and *Videos 1–5*). To further determine at what cell cycle stage CLK1 activity is required, we inhibited CLK1 using TG003 at different time points after early S phase release. Consistent with the imaging data, both control and TG003-treated cells entered mitosis normally, as measured by 4N DNA content. However, inhibition of CLK1 before late S-phase impaired progression through mitosis, whereas cells treated 5 hr after early S phase release underwent a round of normal mitotic division, although failed to enter the next cell cycle (*Figure 4D* and *Figure 4—figure supplement 1D*). These results suggest that the primary defects caused by CLK1 inhibition occur in late S-phase and G2 phase, which is when CLK1 levels normally begin to rise (*Figure 2A*). This conclusion is further supported by the observation that CLK1-dependent AS targets, such as those detected in *HMMR* and *CENPE,* are periodically expressed during cell cycle and peak during G2 and M phase (*Figure 4—figure supplement 1E*). Taken together with the earlier results, these data support an important and multifaceted role for CLK1 in the control of cell cycle progression through its function in the global regulation of periodic AS.

## CLK1 expression and CLK1-regulated AS is altered in kidney cancer

The importance of CLK1 for faithful progression through the cell cycle suggests it may play a role the control of cell proliferation in cancer. Supporting this, shRNA knockdown or chemical inhibition of CLK1 with TG003 or KHCB-19 in HeLa cells results in a near complete block in cell proliferation, as measured by anchorage-dependent and -independent colony formation assays (*Figure 4E* and *Figure 4—figure supplement 1F*). As mentioned above, CLK1 is likely the primary target of inhibition in these experiments since RNAi of CLK1 recapitulates the phenotype seen with these chemical inhibitors.

Using RNA-Seq data from the Cancer Genome Atlas (TCGA) (*Cancer Genome Atlas Network, 2013*) we observe that CLK1 displays significantly higher expression in 72 kidney tumors compared to matched normal tissue samples (*Figure 4F*, p=10$^{-5}$, Kolmogorov-Smirnov test). Consistently, most CLK1-controlled AS events that are altered in tumors have expected splicing changes (*Figure 4—figure supplement 2A*). Furthermore, patients with tumors that

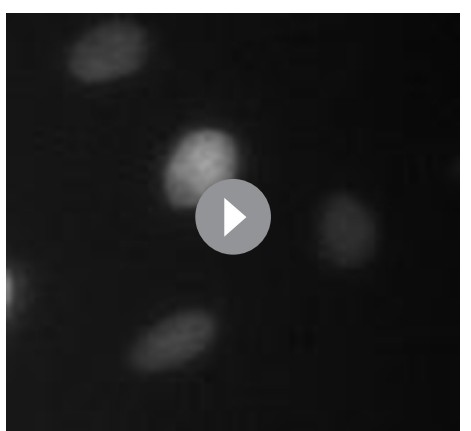

**Video 5.** Zoom of live-cell imaging of TG003-treated HeLa cells. Data associated with *Figure 4*.

have elevated CLK1 expression (i.e. the upper quartile of all samples) have significantly lower survival rates relative to other patients in the comparison group (*Figure 4G*, p=0.007). While there was also an increase in CLK2, CLK3 and CLK4 mRNA expression in these tumors, CLK1 displayed the highest relative mRNA expression levels compared to CLK2 and CLK3 (*Figure 4—figure supplement 2B*). Levels of the CLK4, which is ~80% identical to CLK1, did not correlate with survival differences despite its increased levels in tumors (*Figure 4—figure supplement 2C*). Consistent with this, CLK1-regulated AS events, as defined by the RNA-Seq analysis in *Figure 3*, were also altered across multiple tumor types, including breast, colon, lung and liver (*Figure 4H* and *Figure 4—figure supplement 2D*). These data are further consistent with a multi-faceted role for CLK1 in regulating cell cycle progression, and also suggest that CLK1 contributes to increased cell proliferation in cancer, at least in part through its role in controlling periodic AS.

## Discussion

Previous studies have shown that splicing and the cell cycle are intimately connected processes. Indeed, cell cycle division (CDC) loci originally defined in *S. cerevisiae,* namely *cdc5* and *cdc40*, were subsequently shown to encode spliceosomal components (*Ben-Yehuda et al., 2000*; *McDonald et al., 1999*). Moreover, genome-wide RNAi screens for new AS regulators of apoptosis genes in human cells revealed that factors involved in cell-cycle control, in addition to RNA processing components, were among the most significantly enriched hits (*Moore et al., 2010*; *Tejedor et al., 2015*). An RNAi screen performed in *Drosophila* cells for genes required for cell-cycle progression identified numerous splicing components (*Björklund et al., 2006*) as well as a *Drosophila* ortholog of CLK kinases, *Darkener of apricot Doa* (*Bettencourt-Dias et al., 2004*). In other studies, negative control of splicing during M phase was shown to be dependent on dephosphorylation of the SR family protein, SRSF10 (*Shin and Manley, 2002*), and the mitotic regulator aurora kinase A (AURKA) was shown to control the AS regulatory activity of SRSF1 (*Moore et al., 2010*). Our study shows for the first time that AS patterns are subject to extensive periodic regulation, in part via a global control mechanism involving cell cycle fluctuations of the SR protein kinase CLK1. At least one likely function of this periodic AS regulation is to control the timing of activation of AURKB (*Figure 1*), as well as of numerous other key cell cycle factors shown here to be subject to periodic AS. The definition of an extensive, periodically-regulated AS program in the present study thus opens the door to understanding the functions of an additional layer of regulation associated with cell cycle control and cancer.

Since many RBPs are found to be periodically expressed (*Figure 4A*), it is likely that other aspects of mRNA metabolism are also coordinated with cell cycle stages. For example, differential degradation could potentially contribute to the observed periodic fluctuation of splice isoforms during cell cycle. The degradation of mRNA is closely linked with alternative polyadenylation, which has emerged as a critical mechanism that controls mRNA translation and stability. Generally, shortened 3' UTRs are found in rapidly dividing cells and more aggressive cancers (*Mayr and Bartel, 2009*). This study identified 94 cases periodic alternative poly-A site usage (data not shown) in genes known to regulate cell cycle and/or proliferation, including *SON, CENPF and EPCAM* (*Ahn et al., 2011*; *Bomont et al., 2005*; *Chaves-Perez et al., 2013*). In addition, many mRNAs have recently been found to be translated in a cell cycle dependent fashion (*Aviner et al., 2015*; *Maslon et al., 2014*; *Stumpf et al., 2013*). Interestingly, a fraction of periodically translated genes are also periodically spliced, including key regulators of cell cycle (e.g., *AURKA, AURKB, TTBK1* and *DICER1*). This observation is consistent with recent findings that the regulation of AS and translation may be coupled (*Sterne-Weiler et al., 2013*). In summary, we have demonstrated that AS is subject to extensive temporal regulation during the cell cycle in a manner that appears to be highly integrated with orthogonal layers of cell cycle control. These results thus provide a new perspective on cell cycle regulation that should be taken into consideration when studying this fundamental biological process, both in the context of normal physiology and diseases including cancers.

## Materials and methods

### Cell culture and synchronizations

HeLa (a kind gift from J. Trejo), HEK 293T (from ATCC CRL-3216) and A549 (kind gift from W. Kim) cells were maintained in DMEM (Gibco) medium supplemented with 10% FBS (Gibco). All cells were cultured in humidified incubators with 5% $CO_2$. Cell cycle synchronization was adapted from the protocol of Whitfield et al. (*Whitfield et al., 2002*); ~750,000 log phase HeLa cells were plated in 15 cm dishes in complete media and allowed to attach for 16 hr, reaching <30% confluence. Cells were subsequently treated with 2 mM thymidine (Sigma-Aldrich, St. Louis, MO) for a total of 18 hr, washed 2 times with 1xPBS, and supplemented with fresh complete media for 10 hr. 2 mM thymidine was subsequently added for a second block of 18 hr and washed as described previously. Mitotic block was performed by double thymidine arrest (as above) and release in fresh media for 3.5 hr followed by addition of nocodazole 100 μM (Sigma) for 10 hr. G1 block was performed by serum starvation for 72 hr in DMEM containing 0.05% FBS. For RNA-Seq, cells (both adherent and detached) were harvested every 1.5 hr for 30 hr and frozen immediately for purification of total RNA. To block the activity of CLK1, cells were treated with TG003 (Sigma), KHCB-19 (Tocris, Bristol, UK). To block activity of the proteasome cells were treated with MG132 (Sigma). Drugs were-suspended in DMSO and added to growing cultures at the indicated concentrations and times.

### Flow cytometry and cell cycle analysis

Cells were harvested with trypsin treatment, washed 2 times in cold 1xPBS and subsequently fixed in 80% ice cold ethanol for at least 4 hr. Cells were then washed twice with 1xPBS and suspended in propidium iodide/RNase staining buffer (BD Pharmingen, cat # 550825). Cells were analyzed by flow cytometry to count 10,000 cells that satisfied gating criteria. Data collected were analyzed using ModFit software to discern 2N (G1), S-phase, and 4N (G2 and M) composition.

### Mapping and filtering of RNA-Seq data

RNA-Seq reads were mapped to the human genome (build hg19) using the MapSplice informatics tool with default parameters (*Wang et al., 2010*). The mapped reads were further analyzed with Cufflinks to calculate the level of gene expression with FPKM (Fragments Per Kilobase per Million mapped reads) (*Trapnell et al., 2010*). The levels of alternatively spliced isoforms were quantified with MISO (Mixture-of-Isoforms) probabilistic framework (*Katz et al., 2010*) using the annotated AS events for human hg19version 2. The levels of alternatively spliced isoforms were also quantified with VAST-TOOLS using the event annotation as previously described (*Irimia et al., 2014*). Each AS event was assigned a PSI or PIR value to represent the percent of transcripts with the exon spliced in, or the intron retained, respectively.

### Identification of periodic AS

For identification of periodic AS raw PSI/PIR values were normalized as:

$$normalized\left(\Phi_n^s\right) = \frac{\Phi_n^s - \Phi_{min}^s}{\Phi_{max}^s}$$

where $s$ = 1 to 32,109 for all splicing events; $n$ = 1 to 14 for the 14 samples; $\Phi_{min}$ is the minimum and $\Phi_{max}$ is the maximum PSI value among the 14 samples.

To identify periodic AS events, normalized gene expression values (normalized FPKM values as $e_n$) for the well-known periodic gene, *CCNB1, CCNA2, CCNB2,* and *CENPE*, were used as a starting point to subsequently add curves with broader or sharper peaks as well as shifted to left or right, resulting in 7 periodic expression curves that cover all the phases of cell cycle (*Figure 1—figure supplement 1A*). We term these 'ideal seed curves', which capture intermittent peak times and phase shifts that were not well represented within the initial known periodic genes. To identify genes with similar splicing patterns across the cell cycle, we computed the *EuclideanDistanceED* of each AS event $s$ to the model seed curves $m$ as follows:

$$ED_{m,s} = \sum_{n=1}^{14} \left| normalized(\mathrm{e}_n^m) - normalized(\Phi_n^s) \right|$$

where $m$ = 1 to 7 for all model seed curves, $s$ = 1 to 32,109 for all AS events.

Based on the ranking of distance, a similar cutoff of $ED \leq 2.75$ was set as a minimum requirement for periodic AS. Lastly, we calculated a false discovery rate (FDR) by shuffling PSI values across the 14 time points 10,000 times and calculating how often a random shuffle had a better periodic score than the true periodic score for that event. A maximum FDR of 2.5% was required for a splicing event to be periodic.

## Heat maps, correlations, GO-term analysis, overlap analysis and statistics

Heat maps, hierarchical clustering, and Pearson correlations were generated using GENE-E (www. broadinstitute.org/cancer/software/GENE-E/). All heat maps shown are row-normalized for presentation purposes. Spearman's rank correlation with average linkage was used for clustering. DAVID (http://david.abcc.ncifcrf.gov/gene2gene.jsp) was used for all gene ontology enrichment; terms shown are for biological process (GOTERM_BP_FAT). To test for significance in overlap analysis, overlapping genes in two data sets (i.e. TG003-treatment and periodic AS) and a background set of only co-detected events was used (i.e. genes detected in both experiments). Significance of overlapping gene sets was assessed using the hyper-geometric test. For overlap and correlation analysis between VAST-TOOLS and MISO we used two MISO AS event annotations (HG18 and HG19), due to differences in the input annotation files for these two pipelines. To identify the 4,343 overlapping exons in the heatmap, we used MISO hg19v1 annotations and MISO hg18 annotations, and the VAST-TOOLS annotations. Student's t-test was used to measure significant in cell cycle defects (multi-nucleation and flow-cytometry) as well as semi-quantitative RT-PCR assessment of splice variants. To identify AS events blocked by TG003 or KHCB19, a Student's t-test statistic was used. If a change between G1/S ($t$=0) and G2 ($t$=6) was significant, but not significant in the presence of inhibitors we consider that event to be blocked. For over-representation of periodic introns, we performed Fisher's exact test in a 2x2 contingency table as compared to skipped exons. For differences in expression of CLK1 mRNAs in kidney cancers a Kolmogorov-Smirnov test was performed. For survival differences, the *survdiff* function in the R survival package was used (as discussed in methods below).

## Plasmid construction, transfections and RNAi

The expression constructs were generated by cloning the cDNA of CLK1 into pCDNA3 (for transient expression) or pCDH (for stable transfection) backbones with different epitope tags (HA or Flag) at the N- or C-terminus. The Myc-His-Ubiquitin expression vector is a gift from Dr. Gary Johnson's lab, and the Histone H2B-GFP expression vector is gift from Dr. Angelique Whitehurst's lab. Plasmid transfections were performed using Lipofectamine 2000 (Invitrogen) according to the manufacturer's protocol. The lentiviral vectors of shRNAs were obtained from Addgene in pLKO.1 TRC cloning plasmid through UNC core facility as part of mammalian gene knockdown consortium. Lentiviral infections were performed according to the manufacturer's instruction from System Biosciences (SBI).

## Semi-quantitative RT-PCR assays for monitoring splicing

Cells transfected with shRNA constructs or treated with TG003 were lysed and total RNA was extracted using the Trizol method (Life Technologies). Purified RNA was treated with 1U of RNAse-free DNAase (Promega) for 1 hr at 37°C and reverse transcribed using random hexamer cDNA preparation kit (Applied Biosystem). One-tenth of the RT product was used as the template for PCR amplification (25 cycles of amplification, with a trace amount of Cy5-dCTP in addition to non-fluorescent dNTPs) using gene specific primers listed in *Supplementary file 4*. The resulting gels were scanned with a Typhoon 8600 Imager (GE Healthcare), and analyzed with ImageQuant 5.2 software (Molecular Dynamics/GE Healthcare). Real time PCR was carried out using the SYBR Green kit (Invitrogen) and GAPDH as an internal control.

## Immunoblotting and immunoprecipitation

Proteins were extracted in lysis buffer (CHAPS 1% w/v, 150 mM NaCl, 50 mM $MgCl_2$ with protease inhibitor), resolved by SDS-PAGE and transferred onto PVDF membrane. For immunoprecipitaion experiments to detect ubiquitination, cells were co-transfected with Flag-CLK1 and myc-ubiquitin constructs as above. 36 hr later, TG003 (20 µM) was added for 18 hr. 4 hr prior to harvesting, 10 µM of MG132 was added to the media. Proteins were extracted in lysis buffer as above with the addition of NEM. Incubation with EZ-View FLAG Beads (Sigma) was performed for 2 hr at 4°C. Samples were extensively washed according to the manufacturer's protocol and subjected to immunoblotting. Antibodies and dilutions are listed in *Supplementary file 4*.

## Immunofluorescence and high-content live-cell imaging

For immunofluorescence microscopy, cells were plated on glass coverslips coated with poly-L-Lysine. Cells were then washed twice with 1xPBS, fixed with 4% formaldehyde (Sigma), permeabilized with 0.05% Triton X-100 (Promega) and blocked with 3% BSA (Fisher); all dilutions were made in 1XPBS. For live cell imaging, HeLa cells transduced with Histone H2B-GFP was stably selected as described previously (*Cappell et al., 2010*). Cells were plated in a 6-well format and treated with 2 mM thymidine for 24 hr, subsequently washed and released in fresh complete medium with or without TG003 (20 µM). Cells were imaged using the BD Pathway Microscope with a 10X objective.

## Colony formation assays

HeLa cells stably producing shRNAs targeting CLK1 or control shRNAs were plated at low density (1000 cells/6 $cm^2$) in standard culture medium and allowed to proliferate for 9 days. Cells were then fixed and stained with crystal violet at room temperature. The dried plates were used for estimations of colony diameter and number.

## Kidney cancer analysis of CLK1 and alternative splicing

RNA-Seq data from the The Cancer Genome Atlas (*Ciriello et al., 2013*) was processed as previously described (*Tsai et al., 2015*). Briefly, for mRNA expression, RSEM expression values for the indicated genes were analyzed in 79 paired KIRC (tumor and normal) samples and the paired ks-test was used to test significance. For alternative splicing analysis data from BRCA: Breast invasive carcinoma, COAD: Colorectal adenocarcinoma, KIRC: Kidney renal clear cell carcinoma, LUAD: Lung adenocarcinoma, LUSC: Lung squamous cell carcinoma, LIHC: liver hepatocellular carcinoma were analyzed through the MISO pipeline as described above (*Tsai et al., 2015*). Relapse-free survival was analyzed using Kaplan Meier plots. All plots and statistical analyses (*survdiff*) were generated using the R package version 3.1.1 *survival* function.

## Acknowledgements

We thank Dr. Jan Prins and Darshan Singh for helpful suggestions in analyzing RNA-Seq data, and Drs. Angelique Whitehurst, William Marzluff and Jean Cook for sharing reagents. We thank Drs. Jun Zhu and Ting Ni for advice and assistance in generating strand-specific RNA library. Dr. Rebecca Sinnott provided assistance in live cell imaging experiments. We thank Drs. William Marzluff, Chris Burge, Jean Cook, Michael Emanuele and Mauro Calabrese for helpful comments on the manuscript. This work is supported by an NIH grant (R01CA158283) and Jefferson-Pilot fellowship to ZW, and by grants from the Canadian Institutes of Health Research to BJB. BJB holds the Banbury Chair of Medical Research at the University of Toronto.

## Additional information

### Competing interests

BJB: Reviewing editor, *eLife.*. The other authors declare that no competing interests exist.

## Funding

| Funder | Grant reference number | Author |
|---|---|---|
| National Cancer Institute | R01CA158283 | Zefeng Wang |

The funders had no role in study design, data collection and interpretation, or the decision to submit the work for publication.

## Author contributions

DD, Conceived the project design, Performed the experiments, Analyzed data, Wrote the paper; Y-HT, YW, Analyzed data, Acquisition of data; RW, Analyzed data, Drafting or revising the article; BJB, Analyzed data, Wrote the paper; ZW, Conceived the project design, Analyzed data, Wrote the paper

## Author ORCIDs

Zefeng Wang, http://orcid.org/0000-0002-6605-3637

# Additional files

## Supplementary files

• Supplementary file 1. Periodic splicing events identified during the HeLa cell cycle and associated GO terms. Data associated with *Figure 1*.

• Supplementary file 2. AS events altered by CLK1 inhibition (TG003 10 μM for 16 hr) in 293T cells and associated GO terms. Data associated with *Figure 3*.

• Supplementary file 3. AS events altered by CLK1 inhibition (TG003 10 μM, KHCB19 10 μM) in synchronized HeLa cells. Data associated with *Figure 3*.

• Supplementary file 4. List of PCR primers, shRNA sequences and antibodies used in this study.

## Major datasets

The following dataset was generated:

| Author(s) | Year | Dataset title | Dataset URL | Database, license, and accessibility information |
|---|---|---|---|---|
| Dominguez D, Tsai Y-H, Wang Z | 2016 | An extensive program of periodic alternative splicing linked to cell cycle progression | http://www.ncbi.nlm.nih.gov/geo/query/acc.cgi?acc=GSE81485 | Publicly available at the Gene Expression Omnibus NCBI website (accession no: GSE81485) |

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
