## [Decision Letter]

Thank you for submitting your work entitled "An Extensive Program of Periodic Alternative Splicing Linked to Cell Cycle Progression" for peer review at *eLife*. Your submission has been favorably evaluated by Aviv Regev (Senior editor), a Reviewing editor (Gene Yeo), and three reviewers.

The reviewers have discussed the reviews with one another and the Reviewing editor has drafted this decision to help you prepare a revised submission.

Summary:

The authors identify widespread periodic changes in alternative splicing (AS) (>1000 AS events) that are coordinated with stages of the cell cycle, noticing that periodically regulated intron retention is prominent. The authors discover that SR protein kinase CLK1 is subject to cell cycle-dependent changes and appears to play a central role in the control of AS during cell cycle.

Essential revisions:

1) There were major concerns with regards to the poor overlap of the two pipelines (VAST and MISO) in the identification of AS events. While the reviewers and myself appreciate that the point of the manuscript is not a systematic comparison of both software, the authors are expected to explain the rationale for decisions taken which likely impact the interpretation of the results. Greater detail in the bioinformatics analysis is expected in the revision. Also, why are both tools used? And given the relatively poor overlap, is the use of either or both of these tools justified?

2) A more complete treatment of the contribution of other splicing factors to cell cycle AS is necessary. I do not believe it has to be exhaustive, but in its present form, the manuscript led to concerns by the reviewers that CLK1 is not necessarily a clear outlier (and therefore candidate) in terms of its regulation, compared to other factors. CLK4 modification and levels should be certainly verified by Western blot analysis. Based on these new results (in the revision), the authors are encouraged to alter/broaden their title (as suggested by Reviewer 3 as well).

3) The authors should address the major concerns of Reviewers 1 and 2 with regards to why is the CLK1 inhibitor used (rather than depletion) and address clearly their interpretation of specificity since the inhibitors also target CLK4.

4) The authors should address queries with regards to the PTC-induced degradation of AURKB (Reviewer 1) and known IR events (Reviewer 3).

Reviewer #1:

In this interesting paper, the Blencowe and Wang labs investigate the role of Alternative splicing (AS) during cell cycle progression.

First, Dominguez and co-authors carried out a sequence analysis of the human transcriptome during two continuous cell cycles. This analysis resulted in the identification of widespread periodic changes in AS that are coordinated with specific stages of the cell cycle. In particular, they identified 1,747 AS cell cycle-dependent AS events in 1,293 genes. They found that periodically regulated intron retention is the predominant event.

The authors next focused on the SR protein kinase, CLK1 and found that its expression also fluctuates during the cell cycle. These and other results suggested that CLK1 might have a specific role in regulating cell cycle-dependent AS. Following this, the authors went on to identify endogenous CLK1 AS targets using an RNA-Seq approach. Altogether, these data suggests that temporal regulation of splicing by CLK1 is critical for cell cycle progression. Finally, the authors conclusively show that CLK1 is indeed required for normal mitosis and cell proliferation. The authors put forward a model whereby CLK1 has an essential role in controlling cell cycle-dependent AS.

Defining a central role for CLK1 in the regulation of AS during cell cycle is of importance. The authors also nicely show that the levels of CLK1 are indeed self-regulated and depend on its own catalytic activity. The main targets of CLK1 are SR proteins, yet only in the case of CHEK2 pre-mRNA splicing, it is shown that the effect of CLK1 operates via SRSF1. It would be important to show some evidence that links the role of CLK1 in cell cycle regulated splicing with a particular SR protein/s.

In summary, this is a very good study that represents and important contribution to our understanding of Alternative splicing and cell cycle progression.

Specific comments:

1) In the Abstract and again in paragraph two of the Introduction, the authors refer to periodic regulation of gene function at the levels of transcription, protein modification and degradation, etc.; however, they ignore the role of mRNA translation. There are several papers describing the regulation of mRNA translation during mitosis, which would enrich the Discussion section. Some of these include:

Stumpf et al. (2013) The translational landscape of the mammalian cell cycle. Mol Cell. 2013 Nov 21;52(4):574-82. PubMed PMID: 24120665.

Maslon et al. (2014) The translational landscape of the splicing factor SRSF1 and its role in mitosis. *eLife*. 2014 May 6:e02028. doi: 10.7554/*eLife*.02028. PubMed PMID: 24842991.

2) Is there any overlap in those genes found here to be regulated at the AS level and those previously reported to be regulated by mRNA translation?

3) There seems to be independent regulation of transcriptionally and AS-regulated genes during the cell cycle. Are those genes that show an overlap (133) more likely to be regulated by the kinetic control mode of AS regulation? In other words, are those genes preferentially regulated by the processivity of RNA pol II?

4) On Figure 1, has the PTC-induced degradation of AURKB been experimentally determined?

5) On Figure 3, why is a CLK1 inhibitor preferred to a depletion of CLK1 (either via CRISPR or siRNA approach)?

6) On Figure 3, is the *CENPE* AS also regulated by SRSF1, as is the case with the CHEK2 pre-mRNA?

Reviewer #2:

This manuscript by Dominguez et al., describes a thorough and important analysis of transcriptome changes during the cell cycle, with particular emphasis on alternative splicing. This analysis then motivates the authors to investigate a potential role of the kinase CLK1 in regulating cell cycle splicing and progression, and its possible link to cancer. If fully substantiated, this work would add tremendously to the growing body of knowledge regarding how different cellular growth conditions impact splicing. However, some additional analysis and important controls are required for this manuscript to achieve its full impact.

Firstly, much of the analysis of the RNA-Seq data is poorly defined. Why were VAST-TOOLs and MISO both used and what is the advantage of this? What is "Periodic score" in Figure 1—figure supplement 1? The authors may feel these are readily apparent, but readers should not have to dig deeply through the Methods to understand the first figure. More importantly, Figure 1 should either be replaced with a version that gives raw PSI, or a version with raw PSI should be included in the supplement. Use of the "row normalized" method raises concerns that many of the changes observed in AS events are modest and potentially not of biologic significance.

Secondly, the justification for focusing on CLK1 is not clear. While Figure 3 and Figure 4 certainly support a role of the CLK family in cell cycle splicing, Figure 2 is not convincing that CLK1 is a clear outlier in terms of its regulation. A quick glance through the GE values in supplemental table 1 indicates that genes encoding several SR proteins vary throughout the cell cycle. Have SR proteins other than SRSF1 been analyzed by Western? Moreover, CLK4 should be certainly tested by Western – due to it being a target of TG003 and KHCB-19 (see next point).

Finally, there is significant concern that both the inhibitors used target CLK4 as well as (or more potently than) CLK1. The authors should either conclusively rule out a role for CLK4, or broaden their conclusions to acknowledge a potential role for this family member.

Reviewer #3:

In this work the authors create a genome wide map of oscillating alternative splicing events, and identify CLK1 as a major regulator of a subset of these oscillating events. Roughly, two thirds the paper are focused on exploring the regulatory role of CLK1 through a series of elaborate and thoughtful experiments after establishing it plays a major role in regulating oscillating AS. We found relatively few and mostly minor concerns regarding these later sections. Most critical concerns regard the first, genome wide, analysis.

1) The two pipelines used, based on VAST and MISO, are basically in severe disagreement with respect to which AS events are oscillating. Figure 1—figure supplement 1 shows that not only is the number of events detected very different (244 vs 513) but the overlap of those is a mere 27% or 12%(!). This is coming from two pipelines that use the same downstream procedure to check for oscillation and differ only in their approach to quantify the raw PSI values fed into the pipeline. This poor overlap puts the entire genome wide mapping presented here into questioning. Figure 1—figure supplement 1 may have been added to address this concern (it's not explained) but actually does little to alleviate this concern: Figure 1—figure supplement 1 basically shows that the correlation between VAST and MISO based PSI values are much better for the same condition (diagonal) than for different conditions when looking across the >4K events quantified. That is actually to be expected: most events are either highly included or highly excluded. Such highly included/excluded events would (a) make the extreme values dominating the correlation coefficient and (b) would make the majority of cases that are generally not changing and thus both methods agree on. In fact, a closer look at Figure 1—figure supplement 1 shows the average PSI correlation between VAST and MISO for the same condition (diagonal) is ~0.55 which is again troubling (btw, setting the dark red color for 0.55 is misleading). Some suggestions that may help alleviate the discrepancies between the two pipelines are given below.

Related to this: why does Figure 1—figure supplement 1 contain only 513 events but Figure 1 contains 1747 events? If Figure 1—figure supplement 1 is VAST's list why does Figure 1—figure supplement 1 show 513 for MISO?

2) Basic "normalized PSI" (subheading “Identification of periodic AS”) means we are only looking at *relative* changes of PSI. This may introduce a lot of variability/noise to the analysis and may contribute to the substantial differences between the results from the two analysis pipelines (see above). The authors may be better off screening for events for which (a) min(max(PSI) > VAL1) and (b) max(PSI) – min(PSI) > VAL2. Similarly, increasing the threshold on number of reads per event to be included could help. For example, using VAL1=VAL2 = 20% could help avoid fluctuations that may just appear as periodic or changing dramatically in a relatively small/insignificant range.

Related to this: the authors do not explain how events are screened for coverage across the experiments.

3) The method for detecting the oscillating AS events and its effect on their results is not discussed or evaluated. Specifically, the authors use 7 previously characterized profiles from GE analysis. It's not clear whether the same kind of profiles is the best choice for analyzing periodic AS. For example, profiles 3,4 are similar in phase but one has a wider shape – why would that be the best fit for AS profiles? It seems reasonable to at least compare the results from this approach to an unbiased approach where the entire set of AS events are clustered with no specific periodic profiles and then the most prominent periodic profiles are extracted or executing a more directed search against theoretical periodic profiles.

4) More convincing/exhaustive search of RBPs that may contribute to cell cycle AS:

While there's significant overlap between CLK1 inhibitor AS events and periodic cell cycle events, this only explains a fraction of all the periodic events they find. It would be good to acknowledge at least that there are likely other factors out there that contribute to this program of AS. The authors should discuss how much of the oscillation signal is explained by CLK1. The WB data from Figure 2 is not exhaustive, but it is convincing for those factors tested. It should be straightforward to supplement this analysis by applying their pipelines and RNA-seq data to report known/suspected splicing factors and/or RBPs that change at the level of GE or AS. Any changes found in this analysis could potentially contribute to the large AS program they observe beyond CLK1 regulation.

5) Analysis on IR/AS-PTC introducing events:

Figure 1 suggests that periodic IR can affect transcript expression levels (probably by the introduction of PTCs) in the case of AURKB and some other genes they examined by qRT-PCR that are important for cell cycle progression. However Figure 1 argues little overlap between splicing regulation and mRNA expression levels, save for 133 overlapping events. Are the events in the overlap of periodic AS and periodic gene expression overrepresented for IR like in their example for AURKB and/or other PTC-introducing AS? The same could be done with CLK1 regulated AS and gene expression that is mentioned in the text later (paragraph one, subheading “CLK1 regulates AS events in genes with critical roles in cell cycle control”).

6) Blot for phospho-SR proteins: Figure 2 shows that CLK1 seems to be unique among splicing factors tested in periodic protein expression level changes. An obvious mechanism that likely contributes to the observed periodic splicing changes is altered SR protein phosphorylation, particularly since SRSF1 doesn't change protein expression (Figure 2). Can the authors use a phospho-RS-specific antibody (1H4?) to see if the phosphorylation state of any SR proteins is altered through the cell cycle and, if so, if any of these phosphorylation changes are deadened upon CLK1 inhibition with TG003?

7) Experimentally test known IR event identified in their analysis in CLK1: Figure 2—figure supplement 1 and B show neither CLK1 mRNA levels nor inclusion of exon 4 changes significantly through the cell cycle. Boutz et al. (2014) and others cited within have described regulated intron retention (or "detained introns") of the upstream and downstream introns flanking CLK1 exon 4 that affects the transcript's localization and stability. Since periodic IR is suggested to be a common point of AS regulation through the cell cycle in this paper, showing that these introns are or are not retained across the cell cycle using additional primer sets, as was done for exon 4 skipping in Figure 2—figure supplement 1, could further rule out splicing regulation or add another interesting layer of regulation controlling CLK1 in the context of the cell cycle. Indeed, the MISO analysis in [Supplementary-material SD1-data] calls IR in this region as one of the 1747 periodic splicing events (event: chr2:201726189-201725961:-@chr2:201724469-201724403:-) so this may very well be an additional layer of regulation upstream of the kinase-dependent, ubiquitin-mediated turnover that the authors convincingly demonstrated in Figure 2—figure supplement 1 and Figure 2. Similarly, the authors could test and report results for variations of CLK1's 3'UTR.

---

## [Author Response]

Essential revisions:

*1) There were major concerns with regards to the poor overlap of the two pipelines (VAST and MISO) in the identification of AS events. While the reviewers and myself appreciate that the point of the manuscript is not a systematic comparison of both software, the authors are expected to explain the rationale for decisions taken which likely impact the interpretation of the results. Greater detail in the bioinformatics analysis is expected in the revision. Also, why are both tools used? And given the relatively poor overlap, is the use of either or both of these tools justified?*

The MISO and VAST-TOOLS pipelines utilize different statistical frameworks for the assessment of AS events, which result in different sensitivities for detection of AS events. More importantly, different libraries of annotated AS events are also utilized by these pipelines, such that they capture overlapping but also distinct events. For example, VAST-TOOLS includes annotations for microexons and a larger set of retained introns than MISO. Accordingly, employing the two pipelines provides a more comprehensive survey of periodic AS, whereas a complete overlap in the detected AS events is not expected. Importantly, we observe a strong correlation between PSI values for periodic AS events detected by both pipelines (Spearman’s Rho > 0.8, p value < 10-16). We have clarified these points in paragraph one, subheading “Alternative splicing is coordinated with different cell cycle phases” and Figure 1—figure supplement 1 in the manuscript of the revised manuscript and below in response to Reviewer 2 and Figure 7.

It is also worth noting that raw read data were mapped, filtered and analyzed for AS and gene expression independently using MISO and VAST-TOOLS (see Methods). The fact that we obtain very similar results with both pipelines indicates that our overall results are robust and can be recapitulated using independent pipelines. We therefore respectfully disagree with one of the reviewers that the “poor overlap puts the entire genome wide mapping presented here into questioning.”

2) A more complete treatment of the contribution of other splicing factors to cell cycle AS is necessary. I do not believe it has to be exhaustive, but in its present form, the manuscript led to concerns by the reviewers that CLK1 is not necessarily a clear outlier (and therefore candidate) in terms of its regulation, compared to other factors.

Our study focuses in CLK1 for the following reasons: (1) It shows a periodic expression pattern; (2) It affects a large number of AS events by modifying multiple SR proteins that are key splicing factors; and (3) Its levels are controlled by a self-inhibitory feedback circuit that likely accounts for its periodic regulation. While we fully agree that CLK1 is not the only key regulator of cell cycle dependent splicing, an in depth, comprehensive analyses of the role of other splicing regulators in periodic regulation of AS during the cell cycle is beyond the scope of this study.

Nevertheless, to address the reviewers’ comments we have included a new analysis in our manuscript that systematically examines all annotated RNA-binding proteins for periodic expression patterns that may relate to cell cycle dependent splicing regulation. The results from these new analyses are described on page 7 and 8, summarized in Figure 2—figure supplement 1 and Table 1 in the manuscript, and additional information is provided below for the reviewers’ consideration.

Analysis of our RNA-seq data revealed that 96 RNA binding proteins (RBPs) with periodic mRNA expression, and include RS domain-containing factors like SRSF2, SRSF8, TRA2A and SRSF6 (Figure 5). These 96 RBPs were significantly enriched in the GO term “splicing regulation” (adjusted P = 10-4, Figure 5), indicating that periodic AS is likely controlled by multiple RBPs. Correlations between these RBPs and periodic splicing events were also identified (Figure 5), SRSF2 expression for example, significantly correlated with the splicing of the pattern of a retained intron in the SRSF2 transcript. Further supporting a role for these RBPs in controlling periodic splicing was the identification of RNA motifs bound by a subset of periodically expressed RBPs (Figure 5).

Author response image 1.(**A**) Heat map representation of RNA-bound proteins (RBPs) with periodic expression.Row-normalized FPKM levels are shown. (**B**) GO analyses for the functional enrichment in the periodic RBPs. (**C**) Number of periodic AS events that significantly correlate (Spearman’s Rho > [.75], P <..05) with the expression pattern of each RBP during cell cycle. Expression pattern of two known splicing factors, SRSF2 and ESRP2, is shown in inset. (**D**) Average PSI values of periodic that peak at either G1 (red line) or M phase (blue line) (top panel). k- mer enrichment in periodic exons as judged by Z score and separated by cell cycle phase (y-axis = G1-S and x-axis = G2-M) (bottom panel).**DOI:**
http://dx.doi.org/10.7554/eLife.10288.022

CLK4 modification and levels should be certainly verified by Western blot analysis. Based on these new results (in the revision), the authors are encouraged to alter/broaden their title (as suggested by Reviewer 3 as well).

We have repeatedly attempted to measure the levels of CLK4 by immunoblot. However a reliable antibody to CLK4 is not available.

The reviewers are correct in pointing out that the small molecules utilized in our study likely also inhibit the activity of CLK4. However, shRNA mediated depletion of CLK4 did not reveal significant alteration in cell cycle-associated phenotypes (whereas knockdown of CLK1 did; see below), which is another reason why we focused our efforts on CLK1. Furthermore, in our subsequent kidney tumor analysis, CLK4 expression did not correlate with patient outcomes, whereas CLK1 expression did.

3) The authors should address the major concerns of Reviewers 1 and 2 with regards to why is the CLK1 inhibitor used (rather than depletion) and address clearly their interpretation of specificity since the inhibitors also target CLK4.

The main advantage of employing CLK inhibitors vs. RNAi or CRISPR depletion is that they have rapid effects, which is critical for studying temporal control of splicing as it causes rapid changes in splicing (Figure 3—figure supplement 1). Moreover, due to the self- regulatory feedback of CLK1, its efficient depletion in the context of synchronization experiments was not possible with siRNA pools. As such, we found that the only viable way to assess the activity of the kinase in cell cycle control was to use available chemical inhibitors. Nevertheless, as mentioned above, we show in the manuscript that knockdown of CLK1 using a shRNA phenocopies cell cycle defects observed with both inhibitors, providing evidence that the effects of these inhibitors are due to CLK1 inhibition.

4) The authors should address queries with regards to the PTC-induced degradation of AURKB (Reviewer 1) and known IR events (Reviewer 3).

We carried out additional experiments to address this point. As expected, AURKB intron retention isoforms significantly increase upon treatment with cyclohexamide or an inhibitor of the NMD pathway.

Author response image 2.AURKB splicing isoform with retained intron that contains a pre-mature stop codon is indeed a NMD target.The inhibition of NMD causes accumulation of the intron-containing isoform.**DOI:**
http://dx.doi.org/10.7554/eLife.10288.023

Reviewer #1:

Specific comments:

1) In the Abstract and again in paragraph two of the Introduction, the authors refer to periodic regulation of gene function at the levels of transcription, protein modification and degradation, etc.; however, they ignore the role of mRNA translation. There are several papers describing the regulation of mRNA translation during mitosis, which would enrich the Discussion section. Some of these include:

Stumpf. et al. (2013) The translational landscape of the mammalian cell cycle. Mol Cell. 2013 Nov 21;52(4):574-82. PubMed PMID: 24120665.

Maslon et al. (2014) The translational landscape of the splicing factor SRSF1 and its role in mitosis. eLife. 2014 May 6:e02028. doi: 10.7554/eLife.02028. PubMed PMID: 24842991.

We apologize for this oversight, and have now included this point and cited the references in the new Discussion (paragraph two, Discussion).

2) Is there any overlap in those genes found here to be regulated at the AS level and those previously reported to be regulated by mRNA translation?

See our response above.

3) There seems to be independent regulation of transcriptionally and AS-regulated genes during the cell cycle. Are those genes that show an overlap (133) more likely to be regulated by the kinetic control mode of AS regulation? In other words, are those genes preferentially regulated by the processivity of RNA pol II?

This is a very insightful comment, which raises the possibility that the 133 overlapping genes may be periodically spliced as a result of slower or faster polymerase movement across the gene during transcription. According to the conventional kinetic model, certain genes may be transcribed more rapidly in a specific cell cycle stage, which could potentially modulate AS. Unfortunately, we currently lack measurements for pol II elongation rates in the corresponding genes making it difficult to address the reviewer’s comment at the present time. Related to this question are additional analyses regarding the correlation between a gene’s expression and alternative splicing during cell cycle (please see our response to Reviewer 3, point 3).

4) On Figure 1, has the PTC-induced degradation of AURKB been experimentally determined?

We have addressed this point and included a supplementary figure on this. Please see the essential revision point #4 (and Figure 6) for details.

5) On Figure 3, why is a CLK1 inhibitor preferred to a depletion of CLK1 (either via CRISPR or siRNA approach)?

Please see response above.

6) On Figure 3, is the CENPE AS also regulated by SRSF1, as is the case with the CHEK2 pre-mRNA?

This is an interesting possibility but we feel that it is beyond the scope of the present study.

Reviewer #2:

This manuscript by Dominguez et al., describes a thorough and important analysis of transcriptome changes during the cell cycle, with particular emphasis on alternative splicing. This analysis then motivates the authors to investigate a potential role of the kinase CLK1 in regulating cell cycle splicing and progression, and its possible link to cancer. If fully substantiated, this work would add tremendously to the growing body of knowledge regarding how different cellular growth conditions impact splicing. However, some additional analysis and important controls are required for this manuscript to achieve its full impact.

Firstly, much of the analysis of the RNA-Seq data is poorly defined. Why were VAST-TOOLs and MISO both used and what is the advantage of this? What is "Periodic score" in Figure 1—figure supplement 1? The authors may feel these are readily apparent, but readers should not have to dig deeply through the Methods to understand the first figure. More importantly, Figure 1 should either be replaced with a version that gives raw PSI, or a version with raw PSI should be included in the supplement. Use of the "row normalized" method raises concerns that many of the changes observed in AS events are modest and potentially not of biologic significance.

Please see our response to Dr. Yeo’s essential revision #1. We have included a figure with raw PSI values below as well as added data regarding the methodology of periodic score.

Secondly, the justification for focusing on CLK1 is not clear. While Figure 3 and Figure 4 certainly support a role of the CLK family in cell cycle splicing, Figure 2 is not convincing that CLK1 is a clear outlier in terms of its regulation. A quick glance through the GE values in supplemental table 1 indicates that genes encoding several SR proteins vary throughout the cell cycle. Have SR proteins other than SRSF1 been analyzed by Western? Moreover, CLK4 should be certainly tested by Western – due to it being a target of TG003 and KHCB-19 (see next point).

Finally, there is significant concern that both the inhibitors used target CLK4 as well as (or more potently than) CLK1. The authors should either conclusively rule out a role for CLK4, or broaden their conclusions to acknowledge a potential role for this family member.

Please see our response to the essential revision point #3.

Reviewer #3:

In this work the authors create a genome wide map of oscillating alternative splicing events, and identify CLK1 as a major regulator of a subset of these oscillating events. Roughly, two thirds the paper are focused on exploring the regulatory role of CLK1 through a series of elaborate and thoughtful experiments after establishing it plays a major role in regulating oscillating AS. We found relatively few and mostly minor concerns regarding these later sections. Most critical concerns regard the first, genome wide, analysis.

1) The two pipelines used, based on VAST and MISO, are basically in severe disagreement with respect to which AS events are oscillating. Figure 1—figure supplement 1 shows that not only is the number of events detected very different (244 vs 513) but the overlap of those is a mere 27% or 12%(!). This is coming from two pipelines that use the same downstream procedure to check for oscillation and differ only in their approach to quantify the raw PSI values fed into the pipeline. This poor overlap puts the entire genome wide mapping presented here into questioning. Figure 1—figure supplement 1 may have been added to address this concern (it's not explained) but actually does little to alleviate this concern: Figure 1—figure supplement 1 basically shows that the correlation between VAST and MISO based PSI values are much better for the same condition (diagonal) than for different conditions when looking across the >4K events quantified. That is actually to be expected: most events are either highly included or highly excluded. Such highly included/excluded events would (a) make the extreme values dominating the correlation coefficient and (b) would make the majority of cases that are generally not changing and thus both methods agree on. In fact, a closer look at Figure 1—figure supplement 1 shows the average PSI correlation between VAST and MISO for the same condition (diagonal) is ~0.55 which is again troubling (btw, setting the dark red color for 0.55 is misleading). Some suggestions that may help alleviate the discrepancies between the two pipelines are given below.

Related to this: why does Figure 1—figure supplement 1 contain only 513 events but Figure 1 contains 1747 events? If Figure 1—figure supplement 1 is VAST's list why does Figure 1—figure supplement 1 show 513 for MISO?

We have adjusted the color of the heat map scale in Figure 1—figure supplement 1.

The Figure 1—figure supplement 1 heat map representation of the sample correlations is actually comparing normalized PSI (normalized across cell cycle), which is why higher positive correlations between samples in the cell cycle stage are observed. This was meant to demonstrate that even though the overlap between the two pipelines is not perfect, there is still a significant correlation between these two pipelines. Furthermore, it demonstrates that even when using 4,000 AS events periodicity can still be observed in aggregate.

For clarity we have now included the correlation by raw PSI value (which is much higher >0.8). When low and high PSI values are excluded (0.2 >PSI<0.8) the correlation between VAST-TOOLS and MISO are still positive and very significant (~0.5-0.6 Spearman’s Rho), showing that the positive and significant correlation between our AS measurements are not solely driven by high and low PSI values.

Author response image 3.(**A**) Scatter plot representation of raw PSI values called by VAST- TOOLS and MISO pipelines.(**B**) Heat map representation of possible pair-wise correlation between commonly detected exons by VAST-TOOLS and MISO. Note scale ranges from Spearman’s Rho 0.75-1.0. (**C**) Scatter plot representation of periodic scores of AS events detected by MISO and VAST- TOOLS pipelines. Spearman’s Rho is shown above with a P value of > 2.2e-16. (**D**) Boxplot representation of periodic scores measured by VAST-TOOLS or MISO for events detected for events detected as periodic by the different pipelines (statistical significance was measured by Kolmogorov–Smirnov test, *** = 1e-16).**DOI:**
http://dx.doi.org/10.7554/eLife.10288.024

Furthermore, we found a positive correlation between periodic scores for events detected by both pipelines. In addition, AS events called periodic by one pipeline but missed by the other have significantly lower periodic scores (as determined by either pipeline) than events not identified as periodic in either analysis, suggesting differences in sensitivity (Figure 1).

For more information regarding our rationale for the use of both VAST-TOOLS and MISO see the response to the editor above.

Please see our response to the essential point #1 for the related questions.

*2) Basic "normalized PSI" (subheading “Identification of periodic AS”) means we are only looking at* relative *changes of PSI. This may introduce a lot of variability/noise to the analysis and may contribute to the substantial differences between the results from the two analysis pipelines (see above). The authors may be better off screening for events for which (a) min(max(PSI) > VAL1) and (b) max(PSI) –min(PSI) > VAL2. Similarly, increasing the threshold on number of reads per event to be included could help. For example, using VAL1=VAL2 = 20% could help avoid fluctuations that may just appear as periodic or changing dramatically in a relatively small/insignificant range.*

In the main figures we used relative PSI values for presentation purposes, as it more clearly shows the data and patterns of periodicity. Below we show a heat map representation of raw delta PSI. That is, we are showing (PSI value_i_ – min(PSI val_1-14_)) for each event. This (a) represents the raw data and (b) shows the magnitude of change in raw PSI value that the reviewer has requested.

Author response image 4.Different version of the heat map representation of periodically spliced events.Raw PSI values are shown. Diagram below indicates cell cycle phase. The types of AS events were also indicated.**DOI:**
http://dx.doi.org/10.7554/eLife.10288.025

Related to this: the authors do not explain how events are screened for coverage across the experiments.

We are showing only co-detected events. As stated above different mapping and annotation libraries were used for MISO and VAST-TOOLS.

*3) The method for detecting the oscillating AS events and its effect on their results is not discussed or evaluated. Specifically, the authors use 7 previously characterized profiles from GE analysis. It's not clear whether the same kind of profiles is the best choice for analyzing periodic AS. For example, profiles 3,4 are similar in phase but one has a wider shape* –

*why would that be the best fit for AS profiles? It seems reasonable to at least compare the results from this approach to an unbiased approach where the entire set of AS events are clustered with no specific periodic profiles and then the most prominent periodic profiles are extracted or executing a more directed search against theoretical periodic profiles.*

The difference in width of these curves came from analysis on periodic mRNA expression patterns (please see associated manuscript) where known and clearly periodic mRNAs were missed in our first pass analysis due to wider or narrower shapes expression curves. We dealt with this by including wider and narrower periodic seeds.

We also attempted an unbiased clustering approach for both AS and gene expression (this was the easiest first-pass analysis of the data), this analysis revealed very interesting results. However, we have detected ~30,000 AS events making it very difficult to de-convolute the clusters. We also found it hard to assign a specific “score” to AS events belonging to a cluster. Furthermore, we employed a Fourier transform in some of our preliminary analysis to identify periodic expression patterns as was previously reported by Whitfield et al. (2002), and found similar results. We did feel that the Fourier transform method likely works best when more sampling times are taken along the time course, which was a practical and financial limitation to our sequencing based approach. Below is a heat map representation of hierarchical clustering on all detected events. The bar to the right represents events identified as periodic (in red) by our current analysis.

Author response image 5.Unbiased hierarchical cluster of the PSI values of all AS events during cell cycle.**DOI:**
http://dx.doi.org/10.7554/eLife.10288.026

We appreciate the reviewer’s suggestions and would like to reiterate that we have attempted a variety of approaches to analyze these data and chose a concise presentation style to highlight our main point. We feel that an exhaustive comparison of methods to determine periodic AS/gene expression patterns is beyond the scope of this work.

4) More convincing/exhaustive search of RBPs that may contribute to cell cycle AS:

While there's significant overlap between CLK1 inhibitor AS events and periodic cell cycle events, this only explains a fraction of all the periodic events they find. It would be good to acknowledge at least that there are likely other factors out there that contribute to this program of AS. The authors should discuss how much of the oscillation signal is explained by CLK1. The WB data from Figure 2 is not exhaustive, but it is convincing for those factors tested. It should be straightforward to supplement this analysis by applying their pipelines and RNA-seq data to report known/suspected splicing factors and/or RBPs that change at the level of GE or AS. Any changes found in this analysis could potentially contribute to the large AS program they observe beyond CLK1 regulation.

This comment was addressed by additional data and new Figure 2—figure supplement 1 (Figure 6). Please see the response to essential revision point #2.

5) Analysis on IR/AS-PTC introducing events:

Figure 1 suggests that periodic IR can affect transcript expression levels (probably by the introduction of PTCs) in the case of AURKB and some other genes they examined by qRT-PCR that are important for cell cycle progression. However Figure 1 argues little overlap between splicing regulation and mRNA expression levels, save for 133 overlapping events. Are the events in the overlap of periodic AS and periodic gene expression overrepresented for IR like in their example for AURKB and/or other PTC-introducing AS? The same could be done with CLK1 regulated AS and gene expression that is mentioned in the text later (paragraph one, subheading “CLK1 regulates AS events in genes with critical roles in cell cycle control”).

We have verified that AURKB is indeed an NMD target (essential revision #4). Although the overlap is small, we addressed this point and show a table with the overlapping event types (see our response to the other comment #3 of this same reviewer). As you can see here, there is no specific enrichment for any of the four AS types shown here. As suggested by the reviewer the same overlap analysis was carried out for CLK1-regulated events.

6) Blot for phospho-SR proteins: Figure 2 shows that CLK1 seems to be unique among splicing factors tested in periodic protein expression level changes. An obvious mechanism that likely contributes to the observed periodic splicing changes is altered SR protein phosphorylation, particularly since SRSF1 doesn't change protein expression (Figure 2). Can the authors use a phospho-RS-specific antibody (1H4?) to see if the phosphorylation state of any SR proteins is altered through the cell cycle and, if so, if any of these phosphorylation changes are deadened upon CLK1 inhibition with TG003?

We have attempted this. Unfortunately the results were not consistently reproducible in cell synchrony experiments. Furthermore, we obtained different results when we used mab104 and mab1H4. In addition, conflicting results in terms of changes in the pattern of SR protein phosphorylation during cell cycle were reported previously by Fu and Manley’s labs using the same antibody and cell line.

7) Experimentally test known IR event identified in their analysis in CLK1: Figure 2—figure supplement 1 show neither CLK1 mRNA levels nor inclusion of exon 4 changes significantly through the cell cycle. Boutz et al. (2014) and others cited within have described regulated intron retention (or "detained introns") of the upstream and downstream introns flanking CLK1 exon 4 that affects the transcript's localization and stability. Since periodic IR is suggested to be a common point of AS regulation through the cell cycle in this paper, showing that these introns are or are not retained across the cell cycle using additional primer sets, as was done for exon 4 skipping in Figure 2—figure supplement 1, could further rule out splicing regulation or add another interesting layer of regulation controlling CLK1 in the context of the cell cycle. Indeed, the MISO analysis in [Supplementary-material SD1-data] calls IR in this region as one of the 1747 periodic splicing events (event: chr2:201726189-201725961:-@chr2:201724469-201724403:-) so this may very well be an additional layer of regulation upstream of the kinase-dependent, ubiquitin-mediated turnover that the authors convincingly demonstrated in Figure 2—figure supplement 1 and Figure 2. Similarly, the authors could test and report results for variations of CLK1's 3'UTR.

Our data support a model in which CLK1 levels and activity change during the cell cycle. We agree with this reviewer that intron retention of CLK1 may very well be another form of regulation during cell cycle, but feel that investigating this aspect of CLK1 regulation is beyond the scope of what is already a result-rich manuscript.